# Mesenchymal Stem/Stromal Cells: Immunomodulatory and Bone Regeneration Potential after Tumor Excision in Osteosarcoma Patients

**DOI:** 10.3390/bioengineering10101187

**Published:** 2023-10-13

**Authors:** Max Baron, Philip Drohat, Brooke Crawford, Francis J. Hornicek, Thomas M. Best, Dimitrios Kouroupis

**Affiliations:** 1Department of Orthopedics, UHealth Sports Medicine Institute, Miller School of Medicine, University of Miami, Miami, FL 33146, USA; mcb224@med.miami.edu (M.B.); pdrohat@med.miami.edu (P.D.); txb440@med.miami.edu (T.M.B.); 2Sarcoma Biology Laboratory, Department of Orthopedics, Sylvester Comprehensive Cancer Center, Miller School of Medicine, University of Miami, Miami, FL 33136, USA; bxc859@med.miami.edu (B.C.); fjh21@med.miami.edu (F.J.H.); 3Diabetes Research Institute, Cell Transplant Center, Miller School of Medicine, University of Miami, Miami, FL 33136, USA

**Keywords:** osteosarcoma, bone regeneration, immunomodulation, tumor microenvironment, mesenchymal stem/stromal stem cells

## Abstract

Osteosarcoma (OS) is a type of bone cancer that is derived from primitive mesenchymal cells typically affecting children and young adults. The current standard of treatment is a combination of neoadjuvant chemotherapy and surgical resection of the cancerous bone. Post-resection challenges in bone regeneration arise. To determine the appropriate amount of bone to be removed, preoperative imaging techniques such as bone and CT scans are employed. To prevent local recurrence, the current standard of care suggests maintaining bony and soft tissue margins from 3 to 7 cm beyond the tumor. The amount of bone removed in an OS patient leaves too large of a deficit for bone to form on its own and requires reconstruction with metal implants or allografts. Both methods require the bone to heal, either to the implant or across the allograft junction, often in the setting of marrow-killing chemotherapy. Therefore, the issue of bone regeneration within the surgically resected margins remains an important challenge for the patient, family, and treating providers. Mesenchymal stem/stromal cells (MSCs) are potential agents for enhancing bone regeneration post tumor resection. MSCs, used with scaffolds and growth factors, show promise in fostering bone regeneration in OS cases. We spotlight two MSC types—bone marrow-derived (BM-MSCs) and adipose tissue-derived (ASCs)—highlighting their bone regrowth facilitation and immunomodulatory effects on immune cells like macrophages and T cells, enhancing therapeutic outcomes. The objective of this review is two-fold: review work demonstrating any ability of MSCs to target the deranged immune system in the OS microenvironment, and synthesize the available literature on the use of MSCs as a therapeutic option for stimulating bone regrowth in OS patients post bone resection. When it comes to repairing bone defects, both MB-MSCs and ASCs hold great potential for stimulating bone regeneration. Research has showcased their effectiveness in reconstructing bone defects while maintaining a non-tumorigenic role following wide resection of bone tumors, underscoring their capability to enhance bone healing and regeneration following tumor excisions.

## 1. MSC and the Tumor Microenvironment

### 1.1. Harnessing Macrophages and T Cells in the Tumor Microenvironment for Cancer Elimination

The tumor microenvironment (TME) is a dynamic interplay of various stromal and immune cells that profoundly influences tumor growth, progression, and response to therapies. Among the immune cell populations within the TME, macrophages and T cells have emerged as key players with the potential to eliminate cancer cells (Table 1) [1]. Precise monitoring of the immune environment associated with cancer can provide insight into the stage and prognosis of cancer. During the early stages, NK cells and CD8+ cytotoxic T cells are responsible for the destruction of cancer cells [1]. Sampling the infiltration of the tumor can assess the prognosis of the cancer. A higher level of T cell infiltration is indicative of a better prognosis, whereas a macrophage-predominant infiltration correlates with a worse prognosis [1].

Macrophages make up a significant portion of immune cells that invade developing tumors. They are attracted to the tumor site by signals released by the tumor itself. These tumor-infiltrating macrophages (TAMs) play a crucial role in shaping the tumor microenvironment, which can either support or hinder tumor progression. With appropriate guidance, they can exert powerful antitumor effects by eliminating cancerous cells, impeding the formation of new blood vessels, and reducing fibrosis [2]. In particular, TAMs exhibit plasticity and can display both pro- and anti-tumoral functions depending on their polarization states. TAMs show two phenotypes: pro-inflammatory M1 and anti-inflammatory M2. As the tumor progresses, the TME induces the TAMs to polarize from an M1 to an M2 phenotype, which is pro-tumorigenic [1]. After shifting to the M2 phenotype, these TAMs secrete immunosuppressive cytokines such as IL-10 and TGF-B, angiogenic growth factors such as EGF, and metalloproteinases for extracellular matrix remodeling [1]. In terms of their anti-tumor function, TAMs can induce NR4A1+ patrolling monocytes which have been shown to hinder cancer metastasis, specifically to the lung, which is the most common site of metastasis in OS [3]. Specifically, in OS, high levels of M2 TAMs have been implicated with an increased risk of OS metastasis [4]. However, in primary OS, heterogenous TAMs displaying an intermediate phenotype between M1 and M2 have been associated with anti-metastatic effects [4,5]. Additionally, increased levels of macrophages, due to inflammation after surgical resection of the tumor, result in the increased secretion of inflammatory cytokines, such as tumor necrosis factor-α (TNF-α) and interferon-γ (IFN-γ), which play a role in activating the body’s immune system to promote an anti-tumor response within the body [6].

As previously stated, a high concentration of tumor-infiltrating T cells is correlated with a good prognosis. T cells, including cytotoxic CD8+ T cells, CD4+ T helper 1 (Th1) cells, and CD4+ regulatory T cells (Tregs), intricately modulate immune responses within the TME. CD8+ T cells possess the ability to differentiate into cytotoxic T lymphocytes (CTLs), which play a crucial role in combating tumors [3]. These CTLs utilize specialized granules containing perforin and granzyme, which are released through exocytosis. Through this mechanism, they can perform functions that involve the lysis of tumor cells, thereby conveying a better prognosis in patients with cancer [7]. Additionally, it has been shown that native CD8+ T cells play an immunomodulatory role, more specifically in patients with primary OS [8,9].

As cancer progresses, the TME will evolve to recruit more Tregs (pro-tumorigenic), rather than CD4+ Th1 cells (anti-tumorigenic) [1]. Tregs mediate immunosuppression, and higher levels of tumor-infiltrating Tregs are associated with a less favorable prognosis, while Treg depletion is associated with tumor cell death [1]. Simultaneously, CD4+ Th1 cells contribute to the anti-tumoral response by secreting pro-inflammatory cytokines such as IL-2, TNF-α, and IFN-γ [1,10]. These cytokines have multiple functions, including promoting T cell priming and activation, enhancing CTL cytotoxicity, activating macrophages and natural killer (NK) cells to exert anti-tumoral effects, and increasing the overall presentation of tumor antigens [1]. The presence of tumor-infiltrating CD8+ T cells and the production of Th1 cytokines within tumors have been correlated with favorable prognoses in terms of overall survival and disease-free survival across various malignancies [11,12,13,14].

### 1.2. Immunomodulatory Properties of MSCs

Mesenchymal stem/stromal cells (MSCs) can be obtained from numerous human tissues and can differentiate into a wide variety of cell types [15]. The cross-communication between MSC and cancer cells plays a crucial role in the development of multiple aspects of cancer including tumor growth, metastasis, and epithelial-to-mesenchymal transition [16]. Because these cells play such a crucial role in tumor development, their therapeutic, or anti-tumorigenic, role is being vigorously investigated. Specifically, as MSCs are known to exert immunomodulatory effects, herein we focus on their ability to modulate macrophages’ and T cells’ phenotypes and functionality in vivo. The effective functionality of MSCs in immune regulation relies on their initial activation by a pro-inflammatory microenvironment. This occurs when pro-inflammatory mediators including IFNγ, TNFα, IL-1α, IL-1β, and Toll-like receptor (TLR) ligands such as dsDNA are locally secreted and stimulate them [17,18]. When MSCs are exposed to inflammatory signals, they enhance their production of key immunomodulatory molecules like the tryptophan-degrading enzyme indoleamine 2,3-dioxygenase (IDO) and the prostaglandin E2 (PGE2) [17]. In particular, Waterman et al. demonstrated that TLR-priming of MSCs can lead to MSCs polarization towards an anti-inflammatory immunosuppressive phenotype when primed by TLR3 [17,19]. As noted previously, macrophages rely on specific signaling to play an anti-tumor role as opposed to a pro-tumor role. While CD8 and Th1 CD4+ T cells tend to mostly play an anti-tumor role, they also rely on external signaling to amplify their anti-tumor response in the TME. MSCs have emerged as candidates for immunomodulatory interventions due to their ability to interact with various immune cells and modulate immune responses. The literature has been mixed in terms of the role of MSCs in either positively or negatively modulating the tumor microenvironment [20].

Studies have shown that MSCs have the ability to polarize M1 pro-inflammatory macrophages towards an alternative M2 anti-inflammatory phenotype [21,22,23]. More specifically, exposure of MSCs to PGE2 has been shown to result in the polarization of M1 macrophages to the M2 phenotype [24]. As a result, MSCs have the potential to enhance a pro-tumor microenvironment [25]. MSCs are also capable of secreting factors that impair T cell response and increase the number of Tregs. Furthermore, MSCs are capable of driving CD8+ T cells to a more immunosuppressive phenotype, decreasing the secretion of pro-inflammatory cytokines from Th1 cells, inhibiting CD4+ to Th17 differentiation, and upregulating PD1 expression by Tregs [24].

CD146 expression in MSCs plays a key role in the perivascular niche, skeletogenesis, and hematopoietic support in vivo [26,27,28]. On this basis, our previous data showed that the CD146 signature is correlated with innately higher MSC immunomodulatory and secretory capacities and therapeutic potency in vivo. Specifically, CD146+ BM-MSCs show an increased potency, a robust secretory response, a high immunomodulatory function, and a high therapeutic efficacy. In contrast, CD146- BM-MSCs show a decreased potency, a low secretory response, a low immunomodulatory function, and a low therapeutic efficacy [29]. In another study, CD146+ MSCs exposed to a microenvironment simulating bone injury, demonstrated strong paracrine activity through the upregulation of chemokines, pro-inflammatory and pro-angiogenic, and genes involved in immunomodulation [7]. All these findings suggest that the CD146+ subpopulation is involved in the regulation of various immune responses. Additionally, our previous data showed that extracellular vesicles (EVs) from CD146+ MSCs contain certain miRNAs which play a role in the induction of M2 macrophage polarization, the activation of T cells, and the transcription of anti-inflammatory cytokines [30]. 

In a comparative study, CD106 (VCAM-1) protein expression was evaluated in MSCs from human term placental chorionic villi (CV), umbilical cord (UC), adult bone marrow (BM), and adipose (AD) tissues. The CD106 was most highly expressed in the CV-MSCs, whereas the CD106+ and CD106- subpopulations were evaluated for their biological properties. The CD106+ CV-MSCs had higher expression levels of cytokines and a stronger effect on modulating T helper cells compared to the CD106- subpopulation. Specifically, CD106+ CV-MSCs can inhibit the Th1 polarization of CD4+ T cells and more effectively induce Tregs production compared to the CD106- subpopulation [31]. Overall, the CD106+ was identified as a unique subpopulation of CV-MSCs with immunomodulatory properties. Other MSC subpopulations characterized by specific surface markers (STRO1, CD200, and CD271) have been shown to exert immunomodulatory properties [32]. 

### 1.3. Immunomodulatory Effects of MSCs on TME

It is evident that mesenchymal stem cells play a role in the development of cancer, which raises the following question: can they play a similar role in tumor destruction? It is known that MSCs can exert both stimulatory and inhibitory effects on cancer cell growth, invasion, and metastasis (Table 2) [33]. Numerous factors must be taken into consideration when determining the effects of MSCs on cancer growth, such as the timing of MSCs being introduced to the TME, as well as the method of MSC delivery, which can affect the type of contact MSCs have with cells of the TME [20].

Studies have also looked at co-culturing MSCs with other cell types present in the TME. MSCs/macrophages co-cultures result in the increased expression of CD206 and IL-10, with reduced levels of IL-12 in macrophages [23,34]. Additionally, upon engraftment in tissues characterized by a low concentration of TNF-α and IFN-γ, MSCs transform into a pro-inflammatory MSC1 phenotype. This transition prompts the secretion of numerous inflammatory molecules, including IL-1β, interferon alpha, beta, gamma (IFN-α, IFN-β, IFN-γ), and TNF-α. These released factors subsequently augment the phagocytic capabilities of neutrophils and macrophages, as well as the cytotoxic effects of CTLs and NK cells [23]. In our previous study, it was demonstrated that the exposure of macrophages to endometrium-derived MSC EVs resulted in a polarization to the M2 macrophages phenotype, reduced phagocytic capacity, and decreased secretion of 13 pro-inflammatory molecules [30]. This was also demonstrated in an acute inflammation/fibrosis rat model where the infrapatellar fat-pad-derived (IFP) MSC EVs’ therapeutic treatment caused a polarization of the macrophages towards the anti-inflammatory M2 phenotype [35]. 

Other studies have been able to identify specific pathways in which MSCs can play an anti-tumorigenic role. While the overall picture of why MSCs are anti-tumorigenic is less understood, a recent study showed that MSCs exert their anti-tumorigenic effects via downregulation of the NF-κB, PI3K/Akt and Wnt signaling pathways [36]. Downregulation of the PI3K/Akt pathway was specifically associated with anti-tumor and pro-apoptotic effects [36]. Additionally, other potential anti-tumor effects of MSCs are related to apoptosis via the upregulation of TNF-related apoptosis-inducing ligand (TRAIL), G1 arrest, and expression of tumor suppressor genes [36]. A separate study also investigated MSC-TRAIL, showing that it could inhibit tumor progression in two orthotopic Ewing’s sarcoma mouse models [37]. TRAIL was also shown to reduce metastasis to the lungs in this model. Interestingly, the study identified the downregulation of the Wnt pathway as a mechanism of inhibiting cancer growth [37]. 

Overall, MSCs offer the potential to both activate the immune system and enhance anti-tumor responses, while also having the capability to promote anti-inflammatory responses and, thus, promote pro-tumor responses. Studies have demonstrated that MSCs can promote the activation and cytotoxic function of macrophages and T cells, skewing their polarization towards an anti-tumoral phenotype. Additionally, MSCs can stimulate the expansion and activation of T cells, particularly cytotoxic CD8+ T cells, while suppressing the immunosuppressive function of T regulatory cells (Tregs). However, MSCs have also been shown to promote the proliferation of M2 macrophages and enhance the function of Tregs, promoting a pro-carcinogenic environment. Furthermore, the in vivo functionality of MSCs greatly depends on the type of tissue they derived from and on their ex vivo manipulation (e.g., culture conditions and priming). However, the immunomodulatory properties of MSCs provide a rationale for exploring their application in the post-tumor excision setting to enhance both bone regrowth and the immune-mediated elimination of residual cancer cells and ideally improve therapeutic outcomes [23,38].

**Table 2 bioengineering-10-01187-t002:** MSCs and whether their immunomodulatory properties promote a pro- or anti-tumorigenic environment.

Studies	Findings	Effect on TME
Müller et al. (2021) [24]	Exposure of MSCs to PGE2 polarizes M1 macrophages to the M2 phenotype. MSCs drive CD8+ T-cells to an immunosuppressive phenotype and inhibit pro-inflammatory cytokine secretion from Th1 cells. MSCs inhibit CD4+ to Th17 differentiation and upregulate PD1 expression by Tregs. The effects of MSCs on T-cells and macrophages promote an anti-inflammatory environment conducive to tumor growth and metastasis.	Pro-tumorigenic
Boutilier AJ et al. (2021) [21] Prockop DJ et al. (2012) [22]	MSCs have the ability to shift M1 macrophages towards the M2 phenotype and an anti-inflammatory effect	Pro-tumorigenic
Galland et al. (2020) [33]	Priming MSCs with IFN-γ enhances their antigen-presenting ability by upregulating MHC-I and MHC-II expression. MSCs can deliver anti-neoplastic agents to reduce tumor growth without genetic modification. There are concerns about the unknown effects of the TME on MSCs, as they may be influenced to express more pro-tumorigenic activity.	Mixed
Rivera-Cruz CM et al. (2017) [23]	MSCs shift M1 macrophages to the M2 phenotype and have an anti-inflammatory effect. ASCs increase cytotoxic T cell levels in the tumor microenvironment. ASCs upregulate T lymphocyte proliferation, particularly in the presence of pro-inflammatory cytokines.	Mixed
Harrell et al. (2021) [25]	MSC EVs delivering anti-tumorigenic miRNAs inhibit tumor metastasis, growth, angiogenesis, and invasion.	Anti-tumorigenic
Stamatopoulos et al. (2019) [36]	MSCs control the downregulation of NF-κB, PI3K/Akt and Wnt pathways. Downregulation of the PI3K/Akt pathway is associated with pro-apoptotic effects. MSCs potentially induce apoptosis through the upregulation of TNF-related apoptosis-inducing ligand (TRAIL), G1 arrest, and expression of tumor suppressor genes.	Anti-Tumorigenic
Kim et al. (2009) [34]	Macrophages co-cultured with MSCs showed increased expression of CD206 and IL-10. Macrophages co-cultured with MSCs exhibited reduced levels of IL-12. The co-cultured macrophages demonstrated enhanced phagocytic activity.	Anti-Tumorigenic

## 2. The Application of MSCs after OS Tumor Excision to Promote Bone Regeneration

### 2.1. MSCs Bone Regeneration Properties

MSCs hold great promise for bone regeneration and the treatment of fractures and bone defects. These multipotent cells can differentiate into various cell types, including osteoblasts, chondrocytes, and adipocytes. The osteogenic potential of MSCs plays a crucial role in their capacity to generate bone tissue [39]. Specifically, in vitro and in vivo studies have demonstrated the pivotal role of MSCs in the bone healing process through their potential to enhance osteoinduction and osteogenesis. MSCs can contribute to bone repair and regeneration through several mechanisms, including facilitating cell migration, homing, angiogenesis, response to inflammation, and differentiation [40].

One of the main considerations of using MSCs for bone regeneration in OS patients is the concern regarding the safety profile of MSCs in the TME. Various studies have shown that MSCs’ transplantation/infusion in humans is a safe procedure, showing no indications of malignant alterations [41]. In a study involving 226 patients who received human platelet lysate (hPL)-expanded MSCs for their orthopedic conditions, no malignant transformation was observed during a mean follow-up of 11 months [41,42]. Similarly, Tarte et al. (2010) found that clinical-grade MSCs production, which displayed some chromosomal instability, did not impact their transformation potential both in vitro and in vivo [41,43]. Interestingly, chromosomal abnormalities in culture-expanded MSCs appear to be linked to cell senescence due to extensive passaging, rather than tumorigenesis [41,43,44].

MSCs’ osteogenesis differentiation ability is influenced by various cytokines and growth factors involved in signaling pathways such as TGFβ and WNT [40]. Additionally, matrix metalloproteinases (MMPs) play a critical role in the differentiation of MSCs into osteocytes [40]. Moreover, EVs produced by MSCs have emerged as a novel therapeutic method for bone diseases [40]. EVs can be obtained from MSCs using various methods, including ultracentrifugation, which provides a substantial amount of EVs with consistent purity and reproducibility, allowing for reliable in vitro and in vivo studies [30,36,45,46,47,48]. MSC-EVs promote osteoblast proliferation, differentiation, and bone formation, contributing to bone repair and fracture healing [40,49]. EVs are of particular interest because of their potential application in bone regeneration due to their capacity to transport essential molecular components. Notably, microRNAs (miRNAs) carried within EVs originating from MSCs have been identified as pivotal agents in orchestrating angiogenesis and cellular differentiation towards osteoblasts, two key processes for bone formation. Recent studies offer insight into the significance of miRNAs cargos in EVs derived from BM-MSCs undergoing osteogenic differentiation, such as let-7a, miR-199b, miR-218, miR-148a, miR-135b, miR-203, miR-219, miR-299-5p, and miR-302b [49,50]. These miRNAs are regulators of various signaling pathways, including RNA degradation, mRNA surveillance, and the Wnt signaling pathway, contributing to the facilitation of osteogenesis [49,50]. Additional research emphasizes the prominence of pro-osteogenic miRNAs, such as miR-196a, miR-27a, and miR-206, within BM-MSCs EVs [49,51,52]. Moreover, EVs’ content also encompasses an array of growth factors such as soluble growth factors like VEGF, TGFB1, IL-8, and HGF. These proteins collectively contribute to promoting angiogenic activities by influencing endothelial cell proliferation [49,53,54]. Additionally, miRNAs cargos in MSCs EVs such as miR210, miR126, miR132, and miR21 have been identified as contributors to angiogenesis [49,54].

### 2.2. MSC-Based Strategies for Bone Regeneration In Vivo

MSCs have shown promising results in the field of tissue engineering and regenerative medicine, particularly for bone regeneration and defect repair. The utilization of MSCs in cell-based treatments offers numerous advantages. In particular, ASCs and BM-MSCs possess multipotent properties and the unique ability to differentiate and localize the bone, making them highly suitable for stimulating bone regeneration [39,55]. BM-MSCs and ASCs exhibit similarities in their fibroblast-like morphology, surface markers phenotypic profiles [56], and functional traits [57]. Regarding bone tissue engineering, studies comparing the osteogenic potential of human BM-MSCs and ASCs have yielded mixed results. Some suggest that BM-MSCs exhibit greater osteogenic capacity [58,59,60,61,62], while others propose that ASCs have equal or superior potential [58,63,64,65,66]. In terms of their differences, it has been shown that BM-MSCs can more efficiently differentiate into osteoblasts than their ASCs [39,49]. Additionally, harvesting BM-MSCs is limited by the potential pain and morbidity associated with bone marrow aspiration, and only a small fraction (0.001–0.01%) of harvested bone marrow cells are MSCs [58,67]. Additionally, BM-MSCs can exhibit signs of senescence early in their growth [55,68]. In contrast, it has been shown that ASCs are easier to collect than BM-MSCs [56].

Following bone injury, the bone’s extracellular matrix releases a physiological amount of growth factors which play a vital role in bone repair. Among these, major regulators within the bone remodeling cascade are bone morphogenetic proteins (BMPs), vascular endothelial growth factor (VEGF), fibroblast growth factor (FGF), platelet-derived growth factor (PDGF), transforming growth factor-β1 (TGF-β1), and insulin-like growth factor 1 (IGF-1) [39,69,70,71,72,73,74,75,76]. The migration and homing ability of MSCs to injured sites are crucial initial steps for bone formation and defect repair in MSC-based therapy [40]. Once at the injury site, MSCs can differentiate into osteoblasts, the primary cell responsible for bone formation. These osteoblasts then facilitate the deposition of new bone matrix, subsequently leading to the restoration of the bone’s integrity [39]. This process is triggered by the response of MSCs to inflammatory factors released from the bone fracture site, such as PDGFs and BMPs, which activate BMP-Smad1/5/8 signaling, promoting MSC migration [40,77,78]. The family of BMPs plays a crucial role in inducing osteogenesis. BMP-2, BMP-4, BMP-5, BMP-6, and BMP-7 are vital for bone formation and repair, with applications in open tibial fractures and non-unions [39,69,70,79]. Particularly, BMP-2 and BMP-7 have gained approval for addressing significant bone defects [39,72]. 

Additionally, hypoxia-inducible factor 1-α (HIF-1α) upregulates stromal cell-derived factor-1 (SDF-1) production in damaged bone cells, facilitating MSC migration to the defect site and enhancing bone regeneration [40,77]. A major pathway for the recruitment of MSCs to the fracture site is facilitated by signaling pathways like the SDF1/CXC chemokine receptor (CXCR) 4 axis [39,80,81]. Simultaneously, TGF-β1 acts as a chemotactic agent, homing local MSCs for osteogenic differentiation through the SMAD signaling pathway [39,75]. 

Simultaneously, MSCs contribute to new vessel formation, further enhancing fracture healing [39,82]. VEGF is essential for angiogenesis and osteogenesis during the normal fracture healing process, facilitating the influx of MSCs that differentiate into osteoblasts via their chemotactic effects [39,71,72,83]. Additionally, VEGF prompts MSC migration and triggers osteoblast differentiation through osteotropic growth factors secreted by activated endothelial cells [39,84]. Also, FGF, especially FGF-2, plays a role in angiogenesis and bone regeneration. Studies administering FGF-2 to rat calvarial defects via a collagen sponge have demonstrated augmented osteoblast differentiation, increased blood vessel formation, and higher bone volume generation based on FGF-2 concentrations [39,73]. Additionally, it has been shown that FGF can upregulate VEGF, which contributes to an environment that promotes osteogenesis and angiogenesis [39,85]. Another growth factor significantly influencing angiogenesis and MSCs migration is PDGF, particularly PDGF-BB [39,74]. PDGF-BB attracts pericytes, contributing to vascular stability, and has implications for fracture repair by promoting the differentiation of MSCs into osteoblastic progenitors during active angiogenesis [39,77]. Lastly, IGF-1, abundant in the bone matrix, creates the necessary osteogenic environment for differentiating recruited BM-MSCs into osteoblasts [39,86].

Extensive research has been conducted using MSC therapies in both animal models and patients, employing different methods such as direct MSC injection, seeding MSCs on synthetic scaffolds, utilizing gene-modified MSCs, and employing allogenic MSC application. The literature surrounding the use of genetically modified MSCs as well as MSCs on synthetic scaffolds has shown positive results in terms of promoting osteogenic differentiation of MSCs and bone regrowth in both animal and human studies [87,88]. While there has been less research on non-scaffolded ASCs in bone regrowth, results from some studies have shown that the use of non-scaffolded MSCs can induce bone regrowth in patients with bone lesions or defects [89]. However, it is important to note that only a limited number of these cell-based strategies have been adopted in clinical practice, and none have yet established themselves as the definitive treatment for delayed or non-union of bone or bone regrowth after tumor excision.

Multiple studies have shown that both ASCs and BM-MSCs can promote bone tissue reconstruction in sites of bone loss with and without multifunctional biomaterials (ceramic, biodegradable polymers, and composite materials) that have been designed to guide bone regeneration [39,90,91,92]. Combinations of MSCs with biomaterials offer a promising approach for addressing patient-specific bone defects after trauma and tumor excision. Notably, Giannotti et al. discussed ex vivo, expanded BM-MSCs embedded in autologous fibrin clots for the treatment of upper-limb non-union fractures. At both the short and long term follow-ups at an average time of 6.7 months and 76 months, respectively, all eight of the patients in this study regained the function of their limb with no indications of excessive tissue growth or tumor development. The functional recovery seen through the use of osteogenically differentiated autologous BM-MSCs within fibrin clots shows they can be safely used to heal non-unions [39,93]. Similarly, Lendeckel et al. observed new bone formation and near-total restoration of calvarial continuity through the application of autologous ASCs within fibrin glue and bone graft to address significant calvarial defects [39,92]. Bone marrow aspirate concentrate (BMAC), containing BM-MSCs, also demonstrated efficacy in bone healing of non-unions, outperforming grafts, and minimizing complications in various studies [39,94,95]. Specifically, culture-expanded BM-MSCs on ceramic biomaterials result in the complete fusion of bone segmental defects upon transplantation in vivo with enduring results at follow-up 6–7 years post operatively [39,96]. Additionally, ASCs, in combination with β-tricalcium phosphate and BMP-2, led to osteo-integrated implants and a full defect reconstruction at 36 months in maxillary reconstruction [39,91].

In addition to other biomaterials, hydrogels, with their biocompatibility, bioactivity, biodegradability, and osteoinductive attributes, hold promise in targeting residual tumor areas within bone defects and promoting the natural regeneration of bone at the injury site [97]. These biomimetic hydrogels closely resemble the extracellular matrix (ECM), providing vital support for the proliferation and differentiation of MSCs into osteoblasts, ultimately facilitating the restoration of lost bone tissue [97,98,99,100].

Other studies have investigated the use of both BM-MSCs and ASCs in bone regeneration, specifically in immunocompromised and OS animal models post tumor excision (Table 3). In an immunodeficient mouse model, the subcutaneous implantation of a biodegradable coralline hydroxyapatite–calcium carbonate (CHACC) composite co-cultured with MSCs led to significant osteogenesis and bone formation [101]. Zheng et al. conducted a review in which they discussed the promotion of bone regrowth in patients with OS post tumor resection. They explored a study in which mice were injected with DLM8-luc OS cells [102,103]. Subsequently, the researchers conducted the resection of the tumor upon maturation of the OS model [102,103]. The mice were then separated into three groups: intravenous MSCs, MSCs at the surgical area, and no MSCs. The mice receiving direct injections of MSCs into the OS tumor site exhibited a significant decrease in local recurrence and a hinderance of OS tumor cell growth compared to the other groups, suggesting that MSCs could be promising for enhancing bone healing in patients with lesions after surgery. Additionally, Lee et al. identified a dose-dependent effect of ASCs. They found that low concentrations of injected ASCs into a tumor xenograft led to the inhibition of OS tumor growth, while higher concentrations of injected ASCs led to the stimulation of tumor growth [102,103,104,105]. Another study used an in vitro OS cell culture model and showed that BM-MSCs co-cultured with epigallocatechin gallate (EGCG) demonstrated both chemoprevention and increased osteogenic differentiation, as indicated by a significant increase in markers of osteoblastic differentiation such as the Runt-related transcription factor and the bone gamma-carboxyglutamic acid-containing protein [106]. The results from this study indicated that BM-MSCs, in conjunction with the EGCG scaffold, promoted osteoblast development in the absence of a pro-tumor environment in an OS cell culture, showing promise as a graft material to promote bone formation in patients post tumor resection [106].

In clinical settings, two separate studies investigated the effect of ASCs in treating aseptic non-unions post a wide oncological resection of OS (Table 3). In both studies, the patients were treated with a scaffold-free ASCs. In Vériter et al. study, three osteosarcoma patients received scaffold-free ASCs post wide bone resection. In all three patients at the 54, 27, and 48 months follow-up, respectively, there was an adequate bone regrowth filling in the bone defect, with no evidence of prolonged inflammation or neoplastic growth [89,107]. Also, Dufrane et al. explored the feasibility of using scaffold-free autologous ASCs to treat long bone non-unions due to bone resection for osteosarcoma [89,108]. ASCs were cultured in osteogenic media and combined with demineralized bone matrix to create 3D osteogenic implants in vitro that were successfully transplanted into two patients’ bone non-unions. The results showed that scaffold-free ASCs have the ability to promote osteogenesis without oncological side effects short-term (within 3 months) and long-term (up to 4 years) post-transplantation. In both studies, it was observed over time that all patients experienced favorable general clinical outcomes characterized by successful fracture healing and effective filling of bone defects [89,107,108]. Importantly, there was no incidence of ASCs promoting a pro-tumor TME.

### 2.3. Combinatory Reparative and Anti-Cancer MSCs Effects after OS Tumor Excision 

In general, MSCs and MSC EVs have been shown to exert pro- and anti-tumorigenic effects. Without modulation, the net effect of MSCs appears to be pro-tumorigenic; however, this does not limit their opportunity to be utilized in anti-tumor therapy [33]. Priming MSCs with IFN-γ was shown to transiently upregulate MHC-I and MHC-II expression, enhancing the antigen-presenting ability of MSCs [33]. MSCs can also deliver anti-neoplastic agents, which decrease tumor growth without genetic modification [33]. While this does seem like a promising approach for limiting the size of a cancer, there are concerns. Specifically, the unknown effects of the TME on the MSCs. MSCs could be subverted by the TME to express further pro-tumorigenic activity, which raises concerns about their efficacy and safety [33]. A recent study discussed utilizing MSC-derived EVs to deliver anti-tumorigenic miRNAs to TME. Specifically, human umbilical cord-derived MSC EVs (UC-MSC EVs) contain miRNA-16-5p and miRNA-3940-5p cargos that reduce the metastatic potential of tumor cells. Also, human BM-MSC EVs contain miRNA-4461 cargo that suppresses the invasive properties of tumor cells, and human AT-MSC EVs contain miRNA-15a cargo that causes the apoptosis of tumor cells [25]. Recent studies recognized that miRNAs delivered via MSC EVs were able to inhibit the metastatic potential of tumors, as well as suppress tumor growth, angiogenesis, and invasion [25]. 

It is evident that MSCs show a potent bone regeneration capacity, which raises the following question: can they be used as a combinatory reparative and anti-cancer therapy after OS tumor excision? Both BM-MSCs and ASCs have the ability to take up certain anti-cancer agents using three main mechanisms: transporters, simple diffusion, and endocytosis. BM-MSCs have a high expression of nucleoside transporters, allowing them to effectively take up the anti-cancer agent gemcitabine [109,110]. Lipophilic anti-cancer agents, such as paclitaxel, were taken up by BM-MSCs and ASCs via a simple diffusion [109,111]. ASCs also show increased levels of the endocytosis mediator clathrin, which may play a role in the internalization of drugs via receptor-mediated endocytosis [109,112]. Specifically, the incubation of BM-MSCs with sorafenib also leads to the effective uptake of the agent via endocytosis [109,113]. 

Anti-cancer effects of drug-loaded MSCs are exerted via their conditioned medium (CM). The CM of gemcitabine-loaded BM-MSCs led to the cell cycle arrest of ductal pancreatic adenocarcinoma cells [109,110], whereas the CM of paclitaxel-loaded BM-MSCs completely inhibited the proliferation of prostate cancer, glioblastoma, acute lymphoblastic leukemia, malignant pleural mesothelioma, and multiple myeloma cells [109,114]. The co-transplantation of paclitaxel-loaded BM-MSCs and acute lymphoblastic leukemia into immunodeficient nude mice resulted in the complete inhibition of tumor formation, and the intra-tumoral injection of these drug-loaded MSCs led to a decrease in tumor size. Specifically, the CM of paclitaxel-loaded BM-MSCs inhibited VEGF, a main player in tumor angiogenesis [109,114]. Also, the CM of paclitaxel-loaded ASCs exhibited a dose-dependent effect on Ewing’s sarcoma, prostate cancer, blastoma, neuroblastoma, and acute lymphoblastic leukemia cells. Ewing’s sarcoma was the most sensitive to the CM of paclitaxel-loaded ASCs compared to the other cancer cells [109,114]. Therefore, MSCs can be effectively loaded in vitro with anti-cancer agents and show potent anti-tumorigenic effects in vitro and in vivo. 

Importantly, studies showed that BM-MSCs and ASCs have the ability to retain their capacity for skeletal differentiation even after being exposed to anti-cancer agents. This indicates that they can be favored for the targeted delivery of anti-cancer agents to tumors originating from the bone such as OS [109]. For instance, Nicolay et al. studied the effect of cisplatin on BM-MSCs’ characteristics [109,115] and reported that it does not affect MSCs’ morphology, adhesion, viability, or differentiation potential. Also, typical MSC-related surface markers, CD73, CD90, and CD105, show stable expression levels upon cisplatin exposure [109,115]. Additionally, Liang et al. studied the effects of three anti-cancer agents, cisplatin, camptothecin, and vincristine, on ASCs in vitro. ASCs displayed resistance to all the agents used and fully recovered after treatment [109,116]. Furthermore, their stem cell properties remained intact in vitro following the exposure to these drugs [109,116]. The results from these studies suggest that both BM-MSCs and ASCs could be explored further as a combinatory reparative and anti-cancer treatment option for OS (Figure 1). 

### 2.4. Safety Profile of MSCs in the Context of Osteosarcoma Treatment

Herein, we have focused on post-surgery after bone loss and how MSCs can be used to promote bone regeneration. However, there are safety considerations that need to be addressed when utilizing MSCs in OS TME. In a pioneering study, Xu et al. established an animal model of primary osteosarcoma in nude mice using Saos-2 cells and showed that MSCs infusion via the caudal vein resulted in OS growth and pulmonary metastasis [117]. Importantly, CCL5 was identified as playing an important role in Saos-2 proliferation and migration in vitro, suggesting that the therapeutic use of MSCs may need to be avoided in patients with tumors expressing CCL5 [117]. In another study, the chemokine receptor 4 (CXCR4)-expressing osteosarcoma cell line FM52 was co-infused with MSCs into the caudal vein of nude mice. This resulted in enhanced tumor formation and pulmonary metastasis [118]. Specifically, the tumor growth was CXCR4-mediated and promoted through the VEGF pathway. However, when treated with the CXCR4 inhibitor, AMD3100, the MSCs’ VEGF levels were shown to decrease [118]. Recent studies, using the xenograft mouse model of OS, demonstrated that tumor extracellular vesicle-educated MSCs may promote osteosarcoma progression through IL-6 production and lung metastasis through STAT3 activation in tumor cells [119,120]. Importantly, anti-IL-6 receptor antibodies halted the pro-tumor effects of the tumor extracellular vesicle-educated MSCs [119]. In another study, co-cultures of BM-MSC-derived exosomes with Saos-2, MG-63, and MNNG/HOS OS cells lines led to the increased expression of PVT1, a long, noncoding RNA that contributes to OS growth and metastasis [121]. Also, BM-MSC-derived exosomes may boost osteosarcoma development by promoting oncogenic autophagy in nude mice [122]. However, if the autophagy-related gene 5 is silenced in OS cells, it eliminates the pro-tumor influence of BM-MSC exosomes both in vivo and in vitro [122]. Importantly, studies showed that culturing Saos-2 OS cells in MSC-conditioned media (CM) resulted in the protection of OS cells from drug-induced apoptosis [123]. However, this pathway was dependent on STAT3, so the blockage of STAT3 can lead to the increased sensitivity of chemotherapeutic agents in the setting of MSC-induced resistance [123]. Also, MSC-CM treatment increases the anoikis resistance of Saos2 cells in vitro, whereas exogenous MSC-CM promote the survival and metastasis of Saos2 cells in nude mice [124]. These considerations require further research and evaluation before MSCs are used therapeutically after surgical resection of OS.

## 3. Summary

MSCs have the ability to modulate macrophage polarization and their phagocytic activity, while also inhibiting the activation and cytotoxicity of T cells. However, the function of MSCs is context-dependent and influenced by the MSCs’ tissue origin and the tissue microenvironment that they are administered in during therapy. Additionally, there are safety and efficacy challenges related to MSC manufacturing such as replacing animal-derived media with regulatory-compliant media, ensuring the lack of malignant transformation (genetic stability testing), developing the safest route of delivery (local or systemic), and optimizing MSCs dose (up to 5 × 10^6^ cells/kg of body mass or more). Overall, in the context of bone defect repair, BM-MSCs and ASCs show promise in promoting bone regeneration.

Studies have demonstrated the successful reconstruction of bone defects following tumor excision using MSCs, highlighting their potential for enhancing bone healing and reconstruction after tumor resection. The immunomodulatory properties of MSCs or MSC-EVs, coupled with their regenerative capacity, make them attractive candidates for improving therapeutic outcomes in OS patients. However, it is crucial to further explore the immunomodulatory properties of MSCs in post-tumor excision settings to enhance bone regeneration and immune-mediated elimination of residual cancer cells while avoiding environments that induce the pro-tumor effects of MSCs. For this, recent pioneering studies are investigating the loading of MSCs with anti-cancer drugs for enhanced combinatory reparative and anti-cancer therapeutic outcomes.

In conclusion, MSCs hold promise as a therapeutic option for promoting bone regeneration in OS patients post wide bone resection. Their immunomodulatory properties provide an additional benefit by influencing the TME and potentially inhibiting tumor growth. However, further research is needed to fully understand the mechanisms of action and optimize the delivery methods of MSCs for effective and safe treatment. With continued investigation, MSC-based therapies have the potential to improve the outcomes and quality of life for OS patients. Future research should focus on evaluating the long-term safety and efficacy, optimizing the delivery methods, and unraveling the underlying mechanisms to maximize the benefits of MSC-based therapies while minimizing potential risks.

## Figures and Tables

**Figure 1 bioengineering-10-01187-f001:**
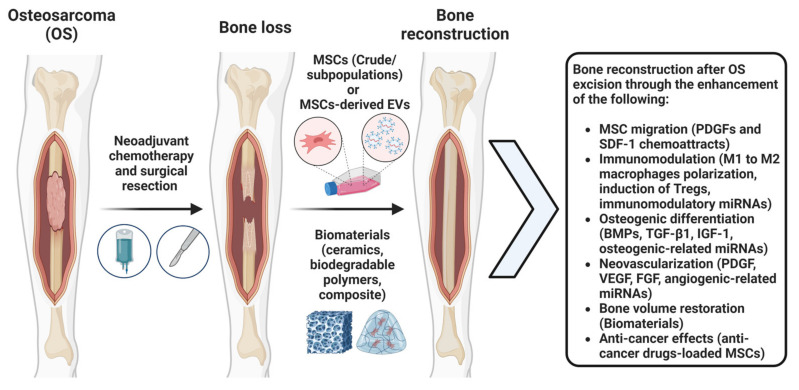
Illustration depicting the dynamic process of bone reconstruction using MSCs, MSCs-derived EVs, and biomaterials following neoadjuvant chemotherapy and surgical resection of OS.

**Table 1 bioengineering-10-01187-t001:** Interplay of M1 macrophages, M2 macrophages, and T cells within the tumor microenvironment.

Cell Type	Role in TME
M1 Macrophages	Have anti-tumoral functions.
	Can induce NR4A1+ patrolling monocytes, which hinder cancer metastasis, specifically to the lung.
	Increased release of inflammatory cytokines (e.g., TNF-α and IFN-γ) can activate the body’s immune system to promote an anti-tumor response.
M2 Macrophages	Can exhibit an anti-inflammatory effect.
	May contribute to a pro-tumor microenvironment.
T Cells	CD8+ T cells can differentiate into cytotoxic T lymphocytes and play a crucial role in combating tumors through the lysis of tumor cells.
	CD4+ Th1 cells secrete proinflammatory cytokines that enhance T-cell priming, activation, CTL cytotoxicity, and anti-tumoral effects.
	CD4+ Regulatory T Cells (Tregs) mediate immunosuppression. Higher levels of tumor-infiltrating Tregs are associated with a less favorable prognosis, while Treg depletion is associated with tumor cell death.
	Tumor-infiltrating CD8+ T cells and Th1 cytokine production correlate with favorable prognoses.

**Table 3 bioengineering-10-01187-t003:** Studies of BM-MSCs and ASCs application for bone regeneration in patients with OS post tumor resection.

BM-MSCs		ASCs	
Studies	Outcomes	Studies	Outcomes
Jo et al. (2023) [106]	BM-MSCs in conjunction with the EGCG scaffold promoted osteoblast development in the absence of a pro-tumor environment.	Smakaj et al. (2022) [89] Vériter et al. (2015) [107] Dufrane et al. (2015) [108]	Examined the use of ASCs in treating aseptic non-unions following a wide oncological resection of OS. Patients treated with scaffold-free adipose-derived grafts showed favorable general clinical outcomes with successful fracture healing and effective filling of bone defects. No promotion of a pro-tumor TME by ASCs was observed.

## Data Availability

Not applicable.

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
