# Peer review of "Mesenchymal Stem/Stromal Cells: Immunomodulatory and Bone Regeneration Potential after Tumor Excision in Osteosarcoma Patients"

_bioengineering, 2023, doi:10.3390/bioengineering10101187_

Round 1

Reviewer 1 Report

This manuscript demonstrated any ability of MSCs to target the deranged immune system in the OS microenvironment and synthesize the available literature on the use of MSCs as a therapeutic option for stimulating bone regrowth in OS patients post bone resection. The review is interesting and well organized. Using MSCs is able to enhancing bone healing and reconstruction after tumor resection. Thus, this manuscript may be accepted after a certain revision. However, the following comments should be addressed before publication.

1. TAMs are critical in promoting tumor progression and therapeutic resistance. In adapting to metabolic changes in the tumor microenvironment (TME), TAMs reprogram their metabolisms and acquire immunosuppressive and pro-tumor properties. How interplay of

TAMs within TME in OS?

2. In the Immunomodulatory effects of MSCs on TME  section, the author demonstrated that MSCs having the capability to promote anti-inflammatory responses and thus promote pro-tumor responses. There are fewer studies related to specific pathways of tissue repair by MSCs, I thus recommend adding the literature of such studies to the increase the innovation of the review.

3. In the MSCs Bone Regeneration Properties   section, the author demonstrated that EVs produced by MSCs have emerged as a novel therapeutic method for bone diseases. Please describe the approach to obtain such stem.

4. Stem cells and biomaterials as a research hotspot in the field of regenerative medicine, the authors were subjected to whether or not it was considered that stem cells combined with hydrogels could be applied to bone regeneration.

5. Citation of the following reference regarding the bone regeneration using Mesenchymal Stem Cells is highly recommended.

Cartilage Regeneration Using Arthroscopic Flushing Fluid-derived Mesenchymal Stem Cells Encapsulated in a One-step Rapid Cross-linked Hydrogel ’. ACTA BIOMATERIALIA, 2018, 79:202-215.

Minor editing of English language required

Author Response

Reviewer 1:

This manuscript demonstrated any ability of MSCs to target the deranged immune system in the OS microenvironment and synthesize the available literature on the use of MSCs as a therapeutic option for stimulating bone regrowth in OS patients post bone resection. The review is interesting and well organized. Using MSCs is able to enhancing bone healing and reconstruction after tumor resection. Thus, this manuscript may be accepted after a certain revision. However, the following comments should be addressed before publication.

 Response: We thank the reviewer for the positive feedback and important remarks. Taking into consideration each of the reviewer’s comments, we believe we have significantly improved our manuscript by correcting typos, clarifying text, and by adding new material/references.

  1. TAMs are critical in promoting tumor progression and therapeutic resistance. In adapting to metabolic changes in the tumor microenvironment (TME), TAMs reprogram their metabolisms and acquire immunosuppressive and pro-tumor properties. How interplay of TAMs within TME in OS? 

Response: We thank the reviewer for this valid comment. We have addressed this point by adding new information at section 1.1 and it reads: ‘Specifically, in OS, high levels of M2 TAMs have been implicated with an increased risk of OS metastasis[4]. However, in primary OS, heterogenous TAMs displaying an intermediate phenotype between M1 and M2 has been associated with anti-metastatic effects[4,5]. Additionally, increased levels of macrophages due to inflammation after surgical resection of the tumor, result in increased secretion of inflammatory cytokines such as tumor necrosis factor-α (TNF-α) and interferon-γ (IFN-γ) that play a role in activating the body’s immune system to promote an anti-tumor response from the body[6].’

  1. In the “Immunomodulatory effects of MSCs on TME “ section, the author demonstrated that MSCs having the capability to promote anti-inflammatory responses and thus promote pro-tumor responses. There are fewer studies related to specific pathways of tissue repair by MSCs, I thus recommend adding the literature of such studies to the increase the innovation of the review.

Response: We thank the reviewer for this valid comment. We have now added new information/literature in section 2.2 and it reads: ‘Regarding bone tissue engineering, studies comparing the osteogenic potential of human BM-MSCs and ASCs have yielded mixed results. Some suggest that BM-MSCs exhibit greater osteogenic capacity [58,59,60,61,62], while others propose that ASCs have equal or superior potential [58,63,64,65,66]. In terms of their differences, it has been shown that BM-MSCs can more efficiently differentiate into osteoblasts than their ASCs[39,49]. Additionally, harvesting BM-MSCs is limited by the potential pain and morbidity associated with bone marrow aspiration, and only a small fraction (0.001–0.01%) of harvested bone marrow cells are MSCs [58,67]. Additionally, BM-MSCs can exhibit signs of senescence early in their growth [55,68]. In contrast, it has been shown that ASCs are easier to collect than BM-MSCs[56].’ Overall, we have significantly reinforced the information provided in our review by supplementing our manuscript with 28 additional references/studies. These references were contributed equally in different sections to support our statements on MSCs immunomodulatory and bone regeneration capacities post-tumor excision in OS patients.

  1. In the “MSCs Bone Regeneration Properties “ section, the author demonstrated that EVs produced by MSCs have emerged as a novel therapeutic method for bone diseases. Please describe the approach to obtain such stem.

Response: We thank the reviewer for this comment. We have now added new text and relevant references to section 2.1 and it reads: ‘EVs can be obtained from MSCs by various methods including ultracentrifugation which provides substantial amount of EVs with consistent purity and reproducibility allowing for reliable in vitro and in vivo studies [30,36,45,46,47,48].’

  1. Stem cells and biomaterials as a research hotspot in the field of regenerative medicine, the authors were subjected to whether or not it was considered that stem cells combined with hydrogels could be applied to bone regeneration.

Response: We thank the reviewer for this comment.  We have added a paragraph about the use of hydrogels in section 2.2. This new paragraph reads: ‘In addition to other biomaterials, hydrogels, with their biocompatibility, bioactivity, biodegradability, and osteoinductive attributes, hold promise in targeting residual tumor areas within bone defects and promoting the natural regeneration of bone at the injury site [97]. These biomimetic hydrogels closely resemble the extracellular matrix (ECM), providing vital support for the proliferation and differentiation of MSCs into osteoblasts, ultimately facilitating the restoration of lost bone tissue [97,98,99,100].’

  1. Citation of the following reference regarding the bone regeneration using Mesenchymal Stem Cells is highly recommended. ‘Cartilage Regeneration Using Arthroscopic Flushing Fluid-derived Mesenchymal Stem Cells Encapsulated in a One-step Rapid Cross-linked Hydrogel ’. ACTA BIOMATERIALIA, 2018, 79:202-215.

Response: We have now added this reference in section 2.2 in paragraph discussing about capacity of MSCs with hydrogel for tissue regeneration.

Reviewer 2 Report

The paper entitled “Mesenchymal Stem/Stromal Cells – Immunomodulatory and Bone Regeneration Potential After Tumor Excision in Osteosarcoma Patients” is an important area of study but it has some limitations which should be incorporated in the final version of the paper.

  1. The paper makes broad statements about the potential of MSCs in bone regeneration and immunomodulation, but it lacks specific details, such as the mechanisms involved or the success rates in actual patients. Providing more specific data and examples would have made the paper more informative.
  2. When discussing potential therapies, it's essential to address safety concerns. The paper should have included a section on the safety profile of MSCs in the context of osteosarcoma treatment, including any potential side effects or complications.
  3. Limited Discussion of Alternative Approaches: The paper mainly focuses on MSCs as a potential therapy. However, it should have discussed alternative approaches to bone regeneration and immunomodulation in osteosarcoma treatment, providing a balanced perspective on different treatment options.
  4. The introduction could benefit from a more detailed explanation of osteosarcoma, its challenges, and the current standard of care before delving into the potential of MSCs.
  5. The paper mentions two types of MSCs (BM-MSCs and ASCs) but doesn't compare their effectiveness or discuss their specific advantages or disadvantages. A comparative analysis would have been informative.
  6. The paper mentions the immunomodulatory effects of MSCs on immune cells like macrophages and T cells but does not go into depth regarding the mechanisms involved. Providing more mechanistic insights would have been beneficial.
  7. The conclusion should summarize the key findings and their implications more explicitly, pointing out the significance of MSCs in osteosarcoma treatment and identifying specific directions for future research.
  8. The paper's abstract mentions "synthesizing available literature," but it does not provide a clear summary of the literature review findings. The abstract should briefly outline the key points from the literature review.
  9. The authors should consider discussing potential challenges and ethical considerations related to the use of MSCs in clinical settings, as this could be relevant to future research and clinical trials.
  10. The paper could benefit from additional references to recent and relevant research articles to support the arguments and claims made in the text.

Author Response

Reviewer 2:

The paper entitled “Mesenchymal Stem/Stromal Cells – Immunomodulatory and Bone Regeneration Potential After Tumor Excision in Osteosarcoma Patients” is an important area of study but it has some limitations which should be incorporated in the final version of the paper.

Response: We thank the reviewer for the positive feedback and important remarks. Taking into consideration each of the reviewer’s comments, we believe we have significantly improved our manuscript by correcting typos, clarifying text, and by adding new material/references.

  1. The paper makes broad statements about the potential of MSCs in bone regeneration and immunomodulation, but it lacks specific details, such as the mechanisms involved or the success rates in actual patients. Providing more specific data and examples would have made the paper more informative.

Response: We thank the reviewer for this valid comment. The focus of the present manuscript was not to elaborate in depth into the MSCs properties as we have already published reviews presenting in depth the bone regeneration and immunomodulation capacity of MSCs. However, herein we have provided the reader with valuable information about MSCs’ bone regeneration and immunomodulatory properties referring to multiple sources, including our previous review article Perez et al (2018) (doi:10.3389/fbioe.2018.00105) and our book chapter Correa D. & Kouroupis D. (2020) Paracrine Effects, Exosomes, and the Secretome. Marx R.E. & Miller R.B. (Eds.), Stem Cells and Regenerative Medicine. pp.57-73 By. Best Publishing Company.

               Specifically, in our previous review article (Perez et al (2018)) we discussed in depth: (1) the processes of endochondral and intramembranous bone formation; (2) the role of MSC as viable building blocks to engineer bone implants; (3) the biomaterials used to direct tissue growth, with a focus on ceramic, biodegradable polymers, and composite materials; (4) the growth factors and molecular signals used to induce differentiation of stem cells into the osteoblastic lineage, which ultimately leads to active bone formation; and (5) the mechanical stimulation protocols used to maintain the integrity of the bone repair and their role in successful cell engraftment. Finally, in the same article we have presented the clinical efficacy of MSC for bone reconstruction in vivo.

               In parallel, herein we are presenting to the reader an overview of the immunomodulatory properties of MSCs’ at section 1.2 by referring among other sources to our book chapter Correa D. & Kouroupis D. (2020) Paracrine Effects, Exosomes, and the Secretome. Marx R.E. & Miller R.B. (Eds.), Stem Cells and Regenerative Medicine. pp.57-73 By. Best Publishing Company. In this book chapter we have performed in depth analysis of MSCs’ immunomodulatory properties and interactions with various immune cell types (T cells, macrophages etc). Herein, we have now added new information at section 1.2 and it reads: ‘The effective functionality of MSCs in immune regulation relies on their initial activation by pro-inflammatory microenvironment. This occurs when pro-inflammatory mediators including IFNγ, TNFα, IL-1α, IL-1β, and Toll-like receptor (TLR) ligands such as dsDNA are locally secreted and stimulate them[17,18]. When MSCs are exposed to inflammatory signals, they enhance their production of key immunomodulatory molecules like the tryptophan-degrading enzyme indoleamine 2,3-dioxygenase (IDO) and prostaglandin E2 (PGE2)[17]. In particular, Waterman et al. demonstrated that TLR-priming of MSCs can lead to MSCs polarization towards an anti-inflammatory immunosuppressive phenotype when primed by TLR3[17,19].’

  1. When discussing potential therapies, it's essential to address safety concerns. The paper should have included a section on the safety profile of MSCs in the context of osteosarcoma treatment, including any potential side effects or complications.

Response: We thank the reviewer for raising this important concern. We have included discussions of the safety of MSCs in different pathologies, as well as added a publication on the safety of non-expanded multipotent stromal cell therapies (2017). We have also included information on the safety of multipotent stromal cell therapies and malignancy from a notable study by Centeno et al. (2011) (10.2174/157488811797904371). Specifically, we have now added a new paragraph discussing MSCs safety in section 2.1. and it reads: ‘One of the main considerations of using MSCs for bone regeneration in OS patients is the concern of safety profile of MSCs in the TME. Various studies have shown that MSCs’ transplantation/infusion in humans is a safe procedure, showing no indications of malignant alterations[41]. In a study involving 226 patients who received human platelet lysate (hPL)-expanded MSCs for orthopedic conditions, no malignant transformation was observed during a mean follow-up of 11 months[41,42]. Similarly, Tarte et al. (2010) found that clinical-grade MSCs production, which displayed some chromosomal instability, did not impact their transformation potential, both in vitro and in vivo[41,43]. Interestingly, chromosomal abnormalities in cultured-expanded MSCs appear to be linked to cell senescence due to extensive passaging, rather than tumorigenesis [41,43,44].’ Also, in Table 3, we have included studies Smakaj et al. (2022) (doi:10.3390/ijms23063057), Veriter et al. (2015) (doi: 10.1371/journal.pone.0139566), and Dufrane et al (2015) (doi: 10.1097/MD.0000000000002220),  which is the only literature looking into utilizing MSCs after osteosarcoma surgery.

  1. Limited Discussion of Alternative Approaches: The paper mainly focuses on MSCs as a potential therapy. However, it should have discussed alternative approaches to bone regeneration and immunomodulation in osteosarcoma treatment, providing a balanced perspective on different treatment options.

Response: We thank the reviewer for this comment. The purpose of this paper was to specifically focus on the utilization of MSCs post surgically rather than describing alternative approaches to bone regeneration and immunomodulation in osteosarcoma treatment. Therefore, we are not discussing immunomodulation with MSCs as a treatment of OS, but rather aimed to provide valuable information on TME, immune cells involved, and anti- or pro-tumorigenic effects of MSCs in OS (section 1). In any case, we have already provided overview on biomaterials used with MSCs (section 2) and heavily referenced within the text our previous already published review on alternative methods for bone regeneration (Perez et al. (doi:10.3389/fbioe.2018.00105))

  1. The introduction could benefit from a more detailed explanation of osteosarcoma, its challenges, and the current standard of care before delving into the potential of MSCs.

Response: We thank the reviewer for this comment. We have now added some more information in the abstract about the standard of care for osteosarcoma and it now reads: ‘Osteosarcoma (OS) is a type of bone cancer that is derived from primitive mesenchymal cells typically affecting children and young adults. The current standard of treatment is a combination of neoadjuvant chemotherapy and surgical resection of the cancerous bone. Post-resection challenges in bone regeneration arise. To determine the appropriate amount of bone to be removed, preoperative imaging techniques such as bone and CT scans are employed. To prevent local recurrence, the current standard of care suggests maintaining bony and soft tissue margins from 3 to 7cm beyond the tumor. The amount of bone re-moved in an OS patient leaves too large of a deficit for bone to form on its own, and requires reconstruction with metal implants or allografts. Both methods require bone to heal, either to the implant or across the allograft junction, often in the setting of marrow-killing chemotherapy. Therefore, the issue of bone regeneration within the surgically resected margins remains an important challenge for the patient, family, and treating providers.’  

  1. The paper mentions two types of MSCs (BM-MSCs and ASCs) but doesn't compare their effectiveness or discuss their specific advantages or disadvantages. A comparative analysis would have been informative.

Response: We thank the reviewer for this valid comment. We have now added a notable study from Mohammad-Ahmed et al. (2018) (doi: 10.1186/s13287-018-0914-1) performing donor-matched comparison of the BM-MSCs and ASCs differentiation capacity. In addition, we have now added multiple references to provide the reader with more information and clarity on their differences and specific advantages and disadvantages (section 2.2). It now reads: ‘Regarding bone tissue engineering, studies comparing the osteogenic potential of human BM-MSCs and ASCs have yielded mixed results. Some suggest that BM-MSCs exhibit greater osteogenic capacity [58,59,60,61,62], while others propose that ASCs have equal or superior potential [58,63,64,65,66]. In terms of their differences, it has been shown that BM-MSCs can more efficiently differentiate into osteoblasts than their ASCs[39,49]. Additionally, harvesting BM-MSCs is limited by the potential pain and morbidity associated with bone marrow aspiration, and only a small fraction (0.001–0.01%) of harvested bone marrow cells are MSCs [58,67]. Additionally, BM-MSCs can exhibit signs of senescence early in their growth [55,68]. In contrast, it has been shown that ASCs are easier to collect than BM-MSCs[56].’

  1. The paper mentions the immunomodulatory effects of MSCs on immune cells like macrophages and T cells but does not go into depth regarding the mechanisms involved. Providing more mechanistic insights would have been beneficial.

Response: We thank the reviewer for this valid comment. We acknowledge that we did not provide and in-depth mechanism of the immunomodulatory effects of MSCs on macrophages and T-cells because this was not the main focus of the manuscript, and we did not want to over present the immunomodulatory properties of MSCs, as we thought this would distract the reader from our main discussion ‘MSCs’ immunomodulatory and bone regeneration potential after tumor excision in OS patients’. In fact, we have already published a thorough analysis of the effects of MSCs on different types of immune cells (T cells, macrophages, B cells, NK cells), and their mediators/mechanisms in our previous published study that we heavily reference in the present manuscript (Correa D. & Kouroupis D. (2020) Paracrine Effects, Exosomes, and the Secretome. Marx R.E. & Miller R.B. (Eds.), Stem Cells and Regenerative Medicine. pp.57-73 By. Best Publishing Company). Importantly, herein the reason we specifically focus on macrophages and T cells is based on their major contribution in the microenvironment during bone regeneration.

  1. The conclusion should summarize the key findings and their implications more explicitly, pointing out the significance of MSCs in osteosarcoma treatment and identifying specific directions for future research.

Response: We thank the reviewer for this valid comment. We have now added new information within the summary related to MSCs safety and efficacy. It reads: ‘Additionally, there are safety and efficacy challenges related to MSC manufacturing such as replacing animal-derived with regulatory-complaint media, ensuring the lack of malignant transformation (genetic stability testing), developing the safest route of delivery (local or systemic), optimizing MSCs dose (up to 5x106 cells/kg of body mass or more).’

  1. The paper's abstract mentions "synthesizing available literature," but it does not provide a clear summary of the literature review findings. The abstract should briefly outline the key points from the literature review.

Response: We thank the reviewer for bringing this point to our attention. We have now added to the abstract a concise summary of the findings from our literature review that it reads: ‘When it comes to repairing bone defects, both MB-MSCs and ASCs hold great potential for stimulating bone regeneration. Research has showcased their effectiveness in recon-structing bone defects while maintaining a non-tumorigenic role following wide resection of bone tumors, underscoring their capability to enhance bone healing and regeneration following tumor excisions’. We hope this helps to better clarify what the reader should expect from our literature review.

  1. The authors should consider discussing potential challenges and ethical considerations related to the use of MSCs in clinical settings, as this could be relevant to future research and clinical trials.

Response: We thank the reviewer for raising this important concern. In general, MSCs application in vivo is not related to any ethical consideration as they are primary cells/progenitors derived from non-embryonic or embryonic-like sources. However, there are potential challenges that are related mainly to their safety and efficacy in vivo.  We have now added relevant discussion in summary section and it reads: ‘Additionally, there are safety and efficacy challenges related to MSC manufacturing such as replacing animal-derived with regulatory-complaint media, ensuring the lack of ma-lignant transformation (genetic stability testing), developing the safest route of delivery (local or systemic), optimizing MSCs dose (up to 5x106 cells/kg of body mass or more).’  

  1. The paper could benefit from additional references to recent and relevant research articles to support the arguments and claims made in the text.

Response: We have significantly reinforced the information provided in our review by supplementing our manuscript with 28 additional references/studies. These references were contributed equally in different sections to support our statements on MSCs immunomodulatory and bone regeneration capacities post-tumor excision in OS patients.

Round 2

Reviewer 1 Report

This manuscript demonstrated any ability of MSCs to target the deranged immune system in the OS microenvironment and synthesize the available literature on the use of MSCs as a therapeutic option for stimulating bone regrowth in OS patients post bone resection. The review is interesting and well organized. Using MSCs is able to enhance bone healing and reconstruction after tumor resection. Furthermore, authors also replied reviewer’s comments properly. Thus, this manuscript can be accepted and published in the journal of Bioengineering.

Minor editing of English language required

Author Response

We would like to thank the reviewer for the valuable comments and for the positive feedback to our edited manuscript version. 

Reviewer 2 Report

Accepted in current form 

Author Response

(The authors gave the same response as above.)
